# IL-21, Inflammatory Cytokines and Hyperpolarized CD8^+^ T Cells Are Central Players in Lupus Immune Pathology

**DOI:** 10.3390/antiox12010181

**Published:** 2023-01-12

**Authors:** Soumya Sengupta, Gargee Bhattacharya, Subhasmita Mohanty, Shubham K. Shaw, Gajendra M. Jogdand, Rohila Jha, Prakash K. Barik, Jyoti R. Parida, Satish Devadas

**Affiliations:** 1Institute of Life Sciences, Bhubaneswar 751023, Odisha, India; 2Regional Centre for Biotechnology (RCB), Faridabad-Gurgaon Expressway, Faridabad 121001, Haryana, India; 3Odisha Arthritis & Rheumatology Centre (OARC), Bhubaneswar 751006, Odisha, India

**Keywords:** SLE, CD8^+^ T cells, inflammatory cytokines, reactive oxygen species, antioxidant, IL-21, STAT’s

## Abstract

Systemic lupus erythematous (SLE) is a chronic autoimmune disorder, broadly characterized by systemic inflammation along with heterogeneous clinical manifestations, severe morbidity, moribund organ failure and eventual mortality. In our study, SLE patients displayed a higher percentage of activated, inflamed and hyper-polarized CD8^+^ T cells, dysregulated CD8^+^ T cell differentiation, significantly elevated serum inflammatory cytokines and higher accumulation of cellular ROS when compared to healthy controls. Importantly, these hyper-inflammatory/hyper-polarized CD8^+^ T cells responded better to an antioxidant than to an oxidant. Terminally differentiated Tc1 cells also showed plasticity upon oxidant/antioxidant treatment, but that was in contrast to the SLE CD8^+^ T cell response. Our studies suggest that the differential phenotype and redox response of SLE CD8^+^ T cells and Tc1 cells could be attributed to their cytokine environs during their respective differentiation and eventual activation environs. The polarization of Tc1 cells with IL-21 drove hyper-cytotoxicity without hyper-polarisation suggesting that the SLE inflammatory cytokine environment could drive the extreme aberrancy in SLE CD8^+^ T cells.

## 1. Introduction

Systemic lupus erythematous (SLE) is a chronic autoimmune disorder, broadly characterized by systemic inflammation along with heterogeneous clinical manifestations. It primarily manifests as a multi organ auto immune disorder, with an inflammatory cytokine milieu, exaggerated autoantibody production and dysfunctional immune response eventually leading to life-threatening situations [1,2]. Among the multiple immune cells involved in precipitating SLE pathogenesis, T cells play a critical role [3]. Previous reports suggest that the CD4^+^ T cell population in SLE is metabolically compromised and is associated with higher ROS generation, mitochondrial hyperpolarization and reduced glutathione, thus dictating severe oxidative stress in these patients [4,5]. However, the role of CD8^+^ T cells in mediating or exacerbating SLE is still unclear; hence, our study focusses on defining these cells, attempting to see if these cells can be plasticized and if these cells can be generated ex vivo [6].

The seminal role of CD8^+^ T cells have been well-studied and established in intracellular infections and in cancers, wherein a robust pro-inflammatory response is crucial for the clearance of pathogens and cancer. While a cytotoxic T cell response with adequate help from helper T cells is exquisitely controlled, uncontrolled pro-inflammatory responses do occur, leading to cytokine storms and cytotoxic-protein-mediated tissue and organ damage [7]. While the role of CD4^+^ T helper cells is clearer, the role of CD8^+^ T cells in immune pathology and most of the autoimmune diseases is limited and is primarily focused on type 1 diabetes and multiple sclerosis [8,9]. With respect to the status of CD8^+^ T cells in SLE, contradictory reports prevail, wherein specific studies suggest hyper-cytotoxicity, whereas others indicate the hypo-cytotoxicity of this population [10,11,12]. Additionally, recent studies suggest the existence of endophenotypes in SLE, based on the cytotoxic and metabolic profiles of CD8^+^ T cells, while the heterogeneous disease profile is quite clearly also dependent on patient’s intrinsic immune profile, age, disease severity and drug regime [13,14]. Our studies attempt to profile these inflammatory CD8^+^ T cells in SLE and to examine if cytokines or other factors drive their aberrancy.

The maturation of a naïve CD8^+^ T cell is critically dependent on the cytokine milieu, apart from the primary antigen and co-stimulus, and is responsible for driving a distinct subtype during their activation. This exclusive and selective cytokine milieu, apart from the primary stimulus, co-stimulation, etc., governs the expression of specific transcription factors, leading to the generation of a specific subtype and the inhibition of others. For example, the cytokine IL-12 is responsible for inducing the expression of transcription factor, Eomesdermin (Eomes), favoring the differentiation of Tc1. Alternatively, cytokine TGF-β and IL-2 is required for the differentiation of T suppressor cells, i.e., CD8^+^ T cells secreting IL-10 [15,16]. Additionally, its differentiation is also regulated by various signaling molecules such as the reactive oxygen species (ROS) and calcium (Ca^2+^) present in the micro environment during CD8^+^ T cell activation. Thus, alteration in the T cell micro environment of these cells can alter phenotypes for, e.g., driving an inflammatory phenotype towards an anti-inflammatory one or reverse, driving lower than optimal activation, driving immune suppression, etc., adversely affecting immune homeostasis and leading to immune pathologies [17,18].

IL-21 is a pleotropic cytokine that has been implicated in the normal function of CD8^+^ T and NK cells during viral infection, enhancing their cytotoxic potential during infection. IL-21 is also responsible for driving T follicular cells (Tfh) differentiation and maturation. In normal physiology, the production of Tfh helps in the maturation of B cells, which is important in maintaining both normal immune homeostasis and protection against various pathogens [19,20].

Among the various factors involved in the pathology of autoimmune diseases, cytokines present in the microenvironment of the cells plays an important role in shaping the immune response. The role of IL-6, IL-12 and IL-10 has been long established in SLE, but in recent studies, the role of IL-21 has come into prominence in SLE pathology. For example, IL-21 is known to upregulate mTOR in Treg cells in SLE patients and diminishes its autophagy and suppressive functions [21]. The indirect targeting of IL-21 by treating patients with belimumab, a specific inhibitor of the soluble B lymphocyte stimulator (BlyS), has shown to decrease serum IL-21, restoring the Treg/Th17 ratio and decreasing disease activity [22]. IL-21 is also a major cytokine that drives Th17 cells in other autoimmune inflammatory diseases like experimental autoimmune encephalomyelitis [23].

The present study essentially focuses on understanding the aberrant and inflammatory SLE CD8^+^ T cell phenotype, their oxidant status, role of redox potential in regulating terminally differentiated cytotoxic T cell and potential application in SLE CD8^+^ T cells. Although previous reports discuss the cytotoxic status of SLE CD8^+^ T cells with certain inconsistencies, others suggest an exhausted population. Additionally, there are studies suggesting oxidative stress in SLE CD8^+^ T cells, but any direct correlation with either their hyper-cytotoxic or exhausted phenotype remains undefined. Consequently, in this study, we report predominantly hyper-cytotoxic, hyper-polarized, higher ROS-secreting CD8^+^ T cells and their oxidant/antioxidant response capabilities. Terminally differentiated human cytotoxic T cells of type 1 (Tc1) also exhibited oxidant/antioxidant sensitivity, validating their plastic nature and suggesting that, like their CD4^+^ T helper counterparts, cytotoxic CD8^+^ T cells are disease-defined and convertible. Additionally, our studies with respect to human Tc1 cells differentiated in the presence of IL-21 named Tc21 also suggest an increased polarization towards an inflammatory phenotype, suggesting that IL-21 may also play an important role in driving inflammatory CD8^+^ T cells in SLE.

## 2. Materials and Methods

### 2.1. Clinical Characteristics of SLE Patients and Demography

For the study, 41 active SLE patients were recruited based on the SLEDAI score. The details of the clinical parameters and medications taken by the patients are given in Table 1. In addition, 26 healthy volunteers devoid of any chronic disease, autoimmunity, infection or malignancies were enrolled for the study. Written informed consent was taken from all of the participants included in the study, and the Institutional Human Ethics Board, Institute of Life Sciences, approved the study. From previous literature and as suggested by the clinician, we have collected samples based on availability. The sample size was determined based on the number of patients visiting the outpatient ward of Odisha Arthritis & Rheumatology Centre (OARC), Bhubaneswar, and as referred by the rheumatologist between November 2021 and July 2022. This study enrolled a total of 52 patients, among which 11 samples had to be excluded based on inferior quality [24].

### 2.2. Whole-Blood and PBMC Phenotyping from SLE Patients and Healthy Controls

Samples of 100 μL whole blood derived from SLE patients and healthy controls were first incubated with Trustain FcX block (BioLegend, San Diego, CA, USA) for 30 min in the dark, followed by the addition of Live/Dead Aqua stain (Invitrogen, Waltham, MA, USA) for 20 min. After the incubation period, a monoclonal antibody cocktail of CD3, CD4, CD8, CXCR5 and ICOS was added again, followed by incubation for 30 min in the dark. Subsequently, RBC lysis was performed, and the cells were treated with BD Fix and lyse. The cells were then washed at 300 g for 8 min, followed by acquisition in Cytoflex S (Beckman Coulter, Brea, CA, USA). To analyze the naïve, effector and memory compartments of CD8^+^ T cells in SLE and HC, CD8^+^ T cells were isolated from PBMCs and then stained with CD45RA and CD45RO, followed by acquisition in BD LSR Fortessa (BD Biosciences, Franklin Lakes, NJ, USA). The details of reagents and used in the experiments are given in Appendix A. All flow cytometric analyses were done in FlowJo software V10.8 (BD Biosciences) [25]. The details of gating strategy are given in Appendix A.

### 2.3. Plasma Cytokine Detection Assay

Neat plasma from SLE and healthy controls were run in duplicates to measure 20 T cell cytokines using a human Milliplex map cytokine assay kit (Millipore, Billerica, MA, USA). The samples were acquired in a Bio-Plex 200 system (Bio-Rad, Hercules, CA, USA), and cytokine concentrations were calculated using the Bio-Plex manager software with a five-parameter (5PL) curve-fitting algorithm applied for standard curve calculation [26].

### 2.4. Differentiation of CD8^+^ T Cells to Tc1 or Tc21 Cells from Healthy Controls

Samples of 5 ml blood were collected from healthy human volunteers, and PBMCs were isolated through Ficoll-dependent density gradient centrifugation. CD8^+^ T cells were then isolated from PBMCs by negative selection using Dyna beads (Invitrogen, MA, USA) according to the manufacturer’s protocol. Cell purity was checked by flow cytometry and was ascertained to be around 85~90% consistently (Appendix A). Isolated cells were cultured in RPMI 1640, supplemented with 10% fetal bovine serum of Australian origin (PAN-Biotech, Aidenbach, Germany), 100 U/mL penicillin, 100 μg/mL streptomycin and 50 mM 2β- mercaptoethanol (Sigma, Burlington, MA, USA). For Tc1 differentiation, 1 × 10^6^ CD8^+^ T cells were activated by Dyna beads human T cell activator CD3/CD28 (Gibco, Dublin, Ireland) according to the manufacturer’s protocol. Additionally, IL-12 (10 ng/mL), IL-2 (100 IU/mL) and anti-IL-4 (10 mg/mL) was added after 24 h. For Tc21 differentiation, IL-21(25 ng/mL) was added along with the above conditions. After 10 days of activation, the cells were then washed with RPMI 1640 media and used for subsequent experiments as indicated [27,28].

### 2.5. Cellular and Mitochondrial ROS from Healthy Controls and SLE CD8^+^ T Cells

Briefly, isolated CD8^+^ T cells were considered for the determination of cellular ROS level. A total of 5×10^5^ cells per ml were stained with 500 nM CellROX Deep Red Reagent (Invitrogen, Waltham, MA, USA) and incubated at 37 °C for 60 min. A total of 1 mM of 1 μL of Sytox Blue was added during the last 15 min of the incubation, to discriminate between live and dead cells. After the incubation, cells were acquired on a Cytoflex S (Beckman Coulter, Brea, CA, USA). To determine mitochondrial hyperpolarization, a total of 1 × 10^6^ cells/mL were stained with 100 nM Mitotracker Red CMXRos (Invitrogen, Waltham, MA, USA) for 15 min at 37 °C. Cells were then washed and acquired on a BD LSR Fortessa using a 561 nm laser for excitation and a 610/20 nm emission filter [14,29]. The details of the gating strategy are given in Appendix A).

### 2.6. Oxidant and Antioxidant Treatment of CD8^+^ T Cells

In brief, 1 × 10^6^ cells/mL of CD8^+^ T cells were seeded in flat-bottomed 24-well plates and were stimulated with 20 ng/mL of PMA (Phorbol 12-myristate 13-acetate, Sigma, Burlington, MA, USA) and 1 μg/mL ionomycin (Sigma, Burlington, MA, USA). For ROS induction, 2.5 μM menadione was added after 30 min of PMA/ionomycin treatment. For the quenching of ROS, 5 mM NAC (N-acetyl-L-cysteine) was added to menadione-treated cells after 15 min of menadione treatment. For the assessment of cytokines in the above conditions, 10 μg/mL of BFA (brefeldin A) was added 2.5 h after PMA/ionomycin treatment [17,30].

### 2.7. Intracellular Staining for Cytokines and Transcription Factors

Around 1 × 10^6^ cells/mL were reactivated with 50 ng/mL PMA and 1 μg/mL ionomycin for 6 h for detecting intracellular cytokines and transcription factor. A total of 10 μg/mL brefeldin A was added to the culture during the last 3 h, and dead cells were excluded using a Zombie Fixable NIR or an Aqua Dye Kit (BioLegend, San Diego, CA, USA). Intracellular cytokine staining was performed using a Cytofix/Cytoperm Fixation/Permeabilization Solution Kit (BD Biosciences, San Jose, CA, USA), while transcription factor staining was performed using a FOXP3 staining buffer set (eBioscience, San Diego, CA, USA), according to the manufacturer’s instructions [17]. The details of the gating strategy are given in Appendix A.

### 2.8. Phospho-STAT Staining for Flow Cytometry

For the intracellular staining of human phospho-STAT3/4, unstimulated or stimulated CD8^+^ T cells were fixed with formaldehyde (final concentration of 1.5%) for 30 min at room temperature. They were then permeabilized with ice-cold methanol with vigorous vortexing and incubated at 4 °C for 24 h. After washing with staining media (PBS with 1% BSA), cells were stained with respective antibodies for 1 h at room temperature and samples acquired on a BD LSR Fortessa [17,31]. The details of the gating strategy are given in Appendix A.

### 2.9. Phospho-mTOR Staining for Flow Cytometry

CD8^+^ T cells from from SLE patients were treated with menadione or menadione and NAC for 30 min at 37 ℃ followed by activation with PMA/ion for 10 mins. The cells were fixed by adding 1.5% formaldehyde for 30 min at room temperature. Then they were permeabilized with ice-cold methanol with vigorous vortexing and incubated at 4 ^o^C for 15 mins. After washing with staining media, cells were stained with p-mTOR (Ser2448) Antibody for 30 min at room temperature and then acquired on a BD LSR Fortessa [30,32]. The details of the gating strategy are given in Appendix A.

### 2.10. Statistics

All flow cytometric analysis was done in FlowJo software V10.8 (BD Biosciences, Franklin Lakes, NJ, USA). Statistical analysis was performed using the GraphPad Prism software, version 9.0.1. Data is presented as mean±standard error of mean (SEM). One-way ANOVA with post hoc Tukey’s multiple comparison test were used to compare statistics among three groups or more. A Mann–Whitney U test was used to compare statistics between HC and SLE patients including the cytokine profile from multiplexing. A paired t test was used to compare Tc1 and Tc21 cells from the same donors. *p* values less than 0.05 were considered significant (* *p* < 0.05, ** *p* < 0.01, *** *p* < 0.001, **** *p* < 0.0001).

## 3. Results

### 3.1. SLE CD8^+^ T Cells Display Higher CXCR5 and ICOS and Altered Compartments

Our primary objective was to examine for altered CD8^+^ T cell populations between healthy volunteers and SLE patients and to report significant differences between them (Figure 1A,B). When we analyzed for the expression of ICOS, a known co-stimulator of CD8^+^ T cells, we found its expression to be significantly higher in SLE patients, strongly indicating their higher activation (Figure 1C,D). Additionally, SLE CD8^+^ T cell expressed higher CXCR5, a chemokine receptor for CXCL13 known to be functionally responsible for B cell maturation (Figure 1E,F). The elevated CXCR5 expression in SLE CD8^+^ T cells suggested their resemblance to Tfh cells and is also functionally evinced in the disease wherein multi-organ auto-antibodies are encountered apart from severe inflammation. Altogether, the augmented expression of ICOS and CXCR5 on CD8^+^ T cells of SLE suggested a pathogenic CD8^+^ T cell population displaying exaggerated cytotoxicity and indirectly capable of inducing higher antibody response via B cell activation.

Since SLE is characterized by very high systemic inflammation and our initial characterization of the SLE CD8^+^ T cell population showed aberrancy, we then looked for known inflammatory cytokines such as IFN-γ and TNF-α, cytotoxic molecules Granzyme B and Perforin C, and transcription factors T-bet and Eomes. Herein, we observed that the co-expression of IFN-γ-Granzyme-B and TNF-α-Perforin C were higher in SLE CD8^+^ T cells (Figure 1I–L), as was the dual expression of transcription factors T-bet and Eomes (Figure 1G,H). Interestingly, the expression of Eomes, Gr B and Perf C (Figure 1M–O) was significantly higher in SLE CD8^+^ T cells, while single positive T-bet (Figure 1P) was lower in SLE, and IL-10 did not show any significant change (Data not shown). The above results indicate that SLE CD8^+^ T cells displayed a significantly more cytotoxic and inflammatory phenotype compared to controls. Interestingly, when we compared between compartments, SLE CD8^+^ T cells had a higher percentages of naïve and memory cells (Figure 1Q,R), which may be due to the higher IL-15 presence in SLE patients. 

### 3.2. SLE Patients Display Higher Pro-Inflammatory Cytokine Profile

To understand systemic inflammation in SLE patients, we assessed 20 T cell cytokines and found that most of the pro-inflammatory ones were elevated in the plasma of SLE patients (Figure 2A), and in direct contrast, only IL-13 among the anti-inflammatories was elevated (Figure 2B). Among the pro-inflammatory cytokines that promote a type I response, IFN-γ, IL-12p40 and TNF-α were significantly elevated in SLE and additionally IL-6, IL-1β, IL-17A, IL-17E and IL-15 were also elevated. IL-21, known to promote inflammation and auto-antibody production, was also elevated in SLE patients as compared to control (Figure 2A). Most of the anti-inflammatory cytokines such as IL-4, IL-10, IL-5 and IL-9 did not show any significant difference in SLE patients compared to healthy controls (Figure 2B). Altogether, this suggested that, despite a strong inflammatory reaction in these patients, the anti-inflammatory response was markedly inadequate or even non-functional. We infer that the significantly higher levels of inflammatory cytokines can profoundly activate cells of both innate and adaptive response, creating a positive feedback loop, in turn exacerbating the pathology of SLE. Most importantly, our results envisage a highly unpredictable and dysregulated immune response due to the diversity and flexibility of the cytokine mix.

### 3.3. SLE CD8^+^ T Cells Display Higher ROS and Mitochondrial Hyperpolarization

Next, we examined the levels of cellular reactive oxygen species (cROS) in SLE CD8^+^ T cells, as ROS is known to maintain the metabolic fitness of T cells and excessive ROS accumulation is known to adversely impact immune responses. In this study, we found increased ROS generation in SLE CD8^+^ T cells (Figure 3A–C), which is in line with previous reports suggesting that ROS was higher in SLE lymphocytes and that the antioxidant glutathione was reduced in these cells. In addition, earlier studies have shown that CD4^+^ T cells in SLE have mitochondrial abnormalities, including increased mitochondrial size and membrane disruption, and led us to look for altered mitochondrial status and the ROS levels of SLE CD8^+^ T cells. Herein, we observed that the mitochondria was also hyperpolarized, which could have resulted in the increased ROS production or vice versa, and consequently or subsequently could have led to an altered redox status in these cells (Figure 3D–F). Altogether, mitochondrial hyperpolarization and the consequent augmented ROS generation and in SLE CD8^+^ T cells suggested a significantly dysregulated redox gradient in them.

### 3.4. Redox Controls Pro- and Anti-Inflammatory Markers in SLE CD8^+^ T Cells

Our mitochondrial and ROS status studies in SLE showed significant aberrancy, and this strongly suggested us to examine their pro- and anti-inflammatory response characteristics. When SLE CD8^+^ T cells were treated with a known oxidant (menadione) or antioxidant (NAC), we found that SLE CD8^+^ T cells responded better to NAC with a significant reduction in IFN-γ production (Figure 4A,G). On the contrary, we did not find any difference in the expression of IL-10 (Figure 4A,F) among untreated cells, cells treated with menadione and.or cells treated with both menadione and NAC. We also report a significant decrease in the expression of Eomes and TNF-α in NAC-treated SLE CD8^+^ T cells when compared to menadione-treated cells (Figure 4D,E) and with the PMA/Ion treated population. This is in corroboration with previous studies, wherein it has been shown that NAC inhibits TNF-α in cells exhibiting higher oxidative stress (Figure 4E,I) and suggested that the restoration of antioxidant potential in the CD8^+^ T cells in SLE reverses the inflammatory phenotype of the CD8^+^ T cells in SLE.

### 3.5. Modulating ROS Reduces p-STAT4 and p-mTOR from SLE CD8^+^ T Cells

ROS is known to upregulate p-STAT3, which consequently led us to examine for the levels of p-STATs, including p-STAT4 in SLE CD8^+^ T cells. We report no significant change in p-STAT3 levels from SLE CD8^+^ T cells (Figure 5A,D) but showed that NAC was able to downregulate activated p-STAT4 levels in SLE CD8^+^ T cells (Figure 5B,E). Our studies indicated that hyper-cytotoxic/polarized SLE CD8^+^ T cells display p-STAT3 that was insensitive to any oxidant/antioxidant treatment, while p-STAT4 was antioxidant-sensitive. Further, this selective sensitivity could be a result of the inflammatory cytokine environs that SLE CD8^+^ T cells exist in or encounter during activation. Crucially, p-STAT3′s insensitivity could be due to the complete uncoupling or desensitization of the IL-10 pathway, and the subsequent inability to modulate IL-10 secretion could be due to the persistent inflammatory environment (Figure 4A,F). On the other hand, the modulation of p-STAT4′s activity with NAC again points to an inflammatory cytokine signaling pathway still responsive to antioxidants. Since mitochondrial hyperpolarization activates mammalian target of rapamycin (mTOR), which can modulate T cell differentiation and cell-death pathways, we analyzed for their oxidant/antioxidant sensitivity in SLE CD8^+^ T cells. p-mTOR from SLE CD8^+^ T cells again was significantly NAC-sensitive (Figure 5C,F) consolidating our earlier conclusion and maybe suggesting upstream presence to p-STAT3 and 4. These results also suggested that p-mTOR could be modulated in SLE CD8^+^ T cells to counter their inflammatory potential. However, p-mTOR’s response with menadione or NAC suggests no correlation with IL-10, while IFN-γ and Eomes had a direct co-relationship with NAC.

### 3.6. Effector Tc1 Cells Exhibit ROS Sensitivity

In order to confirm that the SLE CD8^+^ T cell response was indeed characteristic of an inflammatory cytotoxic T cell, that modulating ROS was indeed capable of regulating the inflammatory/anti-inflammatory response, and that the above results were not artifacts, we treated ex vivo differentiated human Tc1 cells to oxidant/antioxidant reagents. Differentiated Tc1 cells displayed significant IFN-γ production and extremely low IL-10 expression with PMA/Ion activation and reversed the trend with menadione treatment (Figure 6A,B). Not surprisingly, the expression of IFN-γ was restored upon treatment with antioxidant NAC (Figure 6C). Most importantly, our results were similar among the tested batches of terminally differentiated Tc1 cells (Figure 6G–H), wherein the addition of exogenous ROS to Tc1 cells drives an anti-inflammatory response and it was reversed with the addition of an antioxidant. Additionally, Eomesdermin (Eomes), as one of the crucial transcription factors, and TNF-α, as one of the crucial pleiotropic cytokines, which are both required to sustain the cytotoxic and inflammatory response of CD8^+^ T cells, were also examined. When we tested for these proteins in terminally differentiated Tc1 cells with PMA/Ion, we observed the significant expression of Eomes^+^, TNF-α^+^ and double-positive populations (Figure 6D). On menadione treatment, the percentages of both Eomes-positive and dual-positive Eomes and TNF-α population decreased (Figure 6E), suggesting their oxidant sensitivity and on adding NAC dual positivity of Eomes and TNF-α population was again restored indicating NAC can convert these cells to an inflammatory phenotype (Figure 6F). The above results confirmed that both terminally differentiated Tc1 cells, as well as SLE CD8^+^ T cells, were redox-sensitive but were contraindicated in the latter due to the highly inflammatory disease conditions.

### 3.7. ROS Induces Expression of p-STAT3 from Tc1 Cells 

We furthered our studies with Tc1 cells to look for their JAK-STAT pathway perturbations under oxidative stress condition that is known to lead to disruption of cellular homeostasis. Therefore, we analyzed for the phosphorylation status of STAT molecules, when cytokine production was altered through ROS induction. Herein, we found that, following ROS induction through menadione, the switch towards anti-inflammatory cytokine production is accompanied by an increase in p-STAT3 and a reduction in p-STAT4 (Figure 7B,E), wherein it is known that p-STAT3 is known to be required for IL-10, whereas p-STAT4 is required for IFN-γ production. The trend was reversed with antioxidant NAC, causing an increase of p-STAT4 and a decrease in p-STAT3 (Figure 7C,F), and was statistically similar in all of the batches tested (Figure 7G,H). Therefore, our data suggested that ROS regulated the pro- and anti-inflammatory cytokine production by affecting the phosphorylation status of respective STAT molecules in ex vivo differentiated human Tc1 cells. p-STAT3′s modulation in terminally differentiated Tc1 cells gave credence to our hypothesis that hyper-inflammatory cytokine status could desensitize p-STAT3 in SLE CD8^+^ T cells.

### 3.8. IL-21 Polarizes Tc1 Cell towards More Inflammatory Phenotype

We furthered our investigations into understanding how naïve CD8^+^ T cell differentiated into hyper-cytotoxic and inflammatory SLE CD8^+^ T cells. As we found significantly elevated levels of IL-21 from SLE patients, we hypothesized that IL-21 could play a significant role in driving hyper-cytotoxicity, as IL-21 is a pleotropic cytokine known to be pathogenic in SLE by inducing Tfh cells and autoantibody production. Therefore, to examine its effect on CD8^+^ T cells, we differentiated human CD8^+^ T cells into conventional Tc1 and Tc1 in the added presence of IL-21 (Tc21) and not surprisingly found the latter, a highly inflammatory phenotype. A significantly higher percentage of Tc21 cells were dual positive for IFN-γ-Gr B and TNF-α-Perf C when compared to terminally differentiated conventional Tc1 cells (Figure 8A–D). This conclusively proved that IL-21 in SLE could be the major cytokine that drove the hyper-cytotoxic and inflammatory SLE CD8^+^ T cells that we encountered, with Tfh cells as IL-21′s possible cellular source. To further substantiate the role of ROS in Tc21 cells, we then looked for cellular ROS in ex vivo differentiated Tc1 and Tc21 cells. Interestingly we did not find any difference between these two subtypes, indicating that these cells were not metabolically compromised (Figure 8E,F). Most importantly, this suggested that IL-21 supplementation alone, as in our experimental condition, was not sufficient to drive hyper-polarization. Whether extended IL-21 exposure alone could alter CD8^+^ T cell mitochondrial properties or the SLE inflammatory cytokine milieu and disease duration is mandatory for the transformation will remain a clinical and immunological challenge to determine.

## 4. Discussion

We looked for the role of CD8^+^ T cells in SLE, a known chronic autoimmune disorder that manifests with dysregulated immune response and is associated with severe morbidity and mortality without aggressive clinical intervention. The role of aberrant CD4^+^ T cells in SLE has been examined and shown with greater clarity than CD8^+^ T cells, and this study clearly establishes the latter’s role in mediating and sustaining inflammation and cytotoxicity and strongly suggests that they play a seminal role if not being primary in tissue destruction [33,34,35].

We first profiled for SLE CD8^+^ T cells and report a higher expression of activation markers such as ICOS, CXCR5, Eomes and T-bet, inflammatory and cytotoxic molecules such as IFN-γ, TNF-α, Gr-B and Perforin-C [36,37]. We evinced significantly higher Eomes^+^T-bet^+^, IFN-γ^+^Gr-B^+^ and TNF-α^+^Perf-C^+^ cells in SLE, confirming our hypothesis for the role of a “hyper-cytotoxic” CD8^+^T cell phenotype in mediating inflammation. Interestingly we also found that naïve and memory compartments were more numerous in SLE CD8^+^ T cells. While the inflammatory markers lent easy understanding and credence to the disease status, we found the increased number of naïve and memory CD8^+^ T cell compartments to be a bit intriguing and examined the causal cytokines for their elevated levels.

To elaborate on these findings, we did a multiplex analysis of the systemic cytokines present in both healthy volunteers and SLE patients and found that all of the inflammatory cytokines of the panel were elevated. Apart from that, certain cytokines promoting memory T cell formation were elevated significantly, strongly suggesting their possible contribution to the more numerous memory compartment profile seen in SLE patients and may also be responsible for exacerbating the disease pathology through a recall response [38,39,40]. Elevated IL-2, IL-12 and IL-15 also suggest that these cytokines could boost naïve T cell genesis and subsequent activation [41]. While the naïve and memory compartment increase in SLE is explained, we do recognize that the elevated naïve cells could also be T_scm_ stem-cell-like memory cells, as it is well-known that these cells share some overlapping features with the naïve cell population and are reported to be increased in SLE [42]. As we did not analyze for specific markers for stem-cell-like memory cells, we could not delineate the correlation between naïve cells and stem-cell-like memory cells further. However, anti-inflammatory cytokines were also more numerous but did not show any significant difference, suggesting imbalance and a dysregulated anti-inflammatory response. Taken together, these findings suggested that SLE is associated with inflammatory cytotoxic T cells, elevated inflammatory cytokines, cytotoxic proteins and associated cytokines that could drive unwarranted antibody production. All of these aberrations of the immune system profoundly affect the progression of the disease-causing morbidity and eventual mortality if not aggressively treated.

Both we and others have previously reported the seminal role of ROS in mediating and modulating TCR activation, MAPK activity, cytokine response and aberrant T cell response in immune pathology [17,30]. With respect to SLE, these patients have oxidative stress in the peripheral blood lymphocytes and have reduced levels of thiols and glutathione reductase that serve as physiological antioxidants [43]. With respect to delineating the oxidative stress in SLE CD8^+^ T cells, we looked for cellular ROS and mitochondrial hyperpolarization. Although our study essentially confirmed accumulated cellular ROS and mitochondrial hyper-polarization as statistically significant, we were unable to determine the sequential order between the two. Importantly, our data clearly proved that oxidative stress conditions in SLE CD8^+^ T cells, specifically from the mitochondria, had serious implications with respect to immune homeostasis [43,44]. Given that SLE had significantly elevated inflammatory cytokines and that ROS is implicated in inflammation, we directed our studies to try to understand the above factors and their association [14,45,46,47].

With respect to SLE patients, we observed that CD8^+^ T cells responded better with an antioxidant with a significant decrease in IFN-γ, Eomes and TNF-α expression. The above results indicated that elevated ROS and the hyper-polarized mitochondrion evinced in SLE CD8^+^ T cells could be modulated by an antioxidant. This is in line with other studies showing that NAC could plasticize cells towards an inflammatory or anti-inflammatory phenotype, depending upon the initial oxidative stress or disease condition affecting the redox system of the immune cell [30,45,48,49,50,51,52].

We followed up the above findings with studies on STATs to confirm and consolidate that the above findings were indeed pathway-driven [53]. When we analyzed for the status of p-STAT3 and p-STAT4 in SLE CD8^+^ T cells, we observed that p-STAT3 did not show any changes with menadione or NAC treatment and that only p-STAT4 decreased with NAC treatment and is also in line with our previous observation, as p-STAT4 can modulate IFN-γ expression [54]. Since p-mTOR plays an important role in the activation of T cells based on the redox axis, we also wanted to check its status with menadione and NAC treatment in SLE CD8^+^ T cells. We observed that NAC was able to decrease the expression of p-mTOR when compared with both menadione and PMA/Ion-treated cells, indicating that NAC can downregulate oxidative stress in CD8^+^ T cells [45,53]. Overall, our data is in line with a clinical trial study that showed NAC, when given as adjunct therapy in SLE, decreases inflammation and disease activity and also restores the redox axis in T cells by inducing the expression of FOXP3 and reducing p-mTOR [46]. We did not observe any changes in FOXP3 expression (data not shown) due to the fact that our cells were isolated and treated ex vivo, but in the above-mentioned study, NAC was given orally and for a certain period of time and consequently the effect could be due to the mode, dosage and timing of NAC [47].

Although our experiments with SLE CD8^+^ T cells displayed a hyper-cytotoxic, hyper-polarized and hyper-inflammatory phenotype and could be modulated, it led us to ask the question if these were experimental artifacts or a true phenotype of SLE CD8^+^ T cells. Thus, to confirm that terminally differentiated CD8^+^ T cells were truly plastic, we differentiated CD8^+^ T cells towards a Tc1 phenotype and examined their redox sensitivity and consequential plasticity. Surprisingly, we found that menadione decreased the expression of cytokines, transcription factors and JAK-STAT proteins associated with an inflammatory phenotype and promoted the expression of anti-inflammatory cytokines in Tc1. Further, we observed that menadione increased the expression of p-STAT3 and reduced p-STAT4 expression, while the trend reversed with NAC addition. As p-STAT3 and p-STAT4 are known to regulate IL-10 and IFN-γ expression, we hypothesized these to be the key regulators governing the inflammatory/anti-inflammatory axis [54,55,56,57]. We observe that the trends with terminally differentiated Tc1 cells were diametrically opposite to the SLE CD8^+^ T cells and attribute the reversal to the disease process itself. As we discovered significant levels of inflammatory cytokine in SLE patients and also know that the cytokine milieu can dictate activation and phenotype characteristics, we followed this line of thinking. 

While the aberrant SLE CD8^+^ T cell had higher cytotoxic, inflammatory and ROS levels and was suggestive of an inflammatory-cytokine-milieu-driven differentiation, this phenotype was not encountered in conventional Tc1 nor in normal controls. Interestingly, as the SLE plasma displayed higher IL-21 levels and with the known associated antibody pathology, we hypothesized that this cytokine from Tfh cells could be the potential modulator [5]. Thus, we terminally differentiated CD8^+^ T cells from healthy controls in the presence of IL-21 along with IL-12 and IL-2, labeled as Tc21, and compared it to conventional Tc1. We observed that IL-21 did increase the cytotoxic and inflammatory potential of Tc21 cells when compared with Tc1, as was cellular ROS. However, they were not statistically significant, indicating that, although Tc21 had cytotoxic capabilities but did not bear resemblance to the SLE CD8^+^ T cell. Thus, the cytokine milieu, disease stage, immune health, etc., could play a role in transforming the CD8^+^ T cell [58,59]. Although we understand that our study is based on a small sample size, but it essentially addresses the role played by the inflammatory and dysregulated immune environs in plasticizing CD8^+^ T cells. SLE CD8^+^ T cells bear characteristics that can only be partly mimicked in Tc21 cells concreting the above factors. But a study in a large cohort is required to understand which patients can be treated with NAC as an adjunct therapy as heterogeneity and different endophenotypes exist in SLE patients. Targeting IL-21 along with the modulating redox axis of CD8^+^ T cells can also be a viable therapeutic option, as IL-21 is known to promote both Th17 and Tfh in SLE cells, and our study also indicates that priming with IL-21 can polarize CD8^+^ towards an inflammatory phenotype. Additionally, our study essentially throws light into the crucial aspect of the redox axis playing a significant role in determining or modulating the phenotype of SLE CD8^+^ T cells.

## Figures and Tables

**Figure 1 antioxidants-12-00181-f001:**
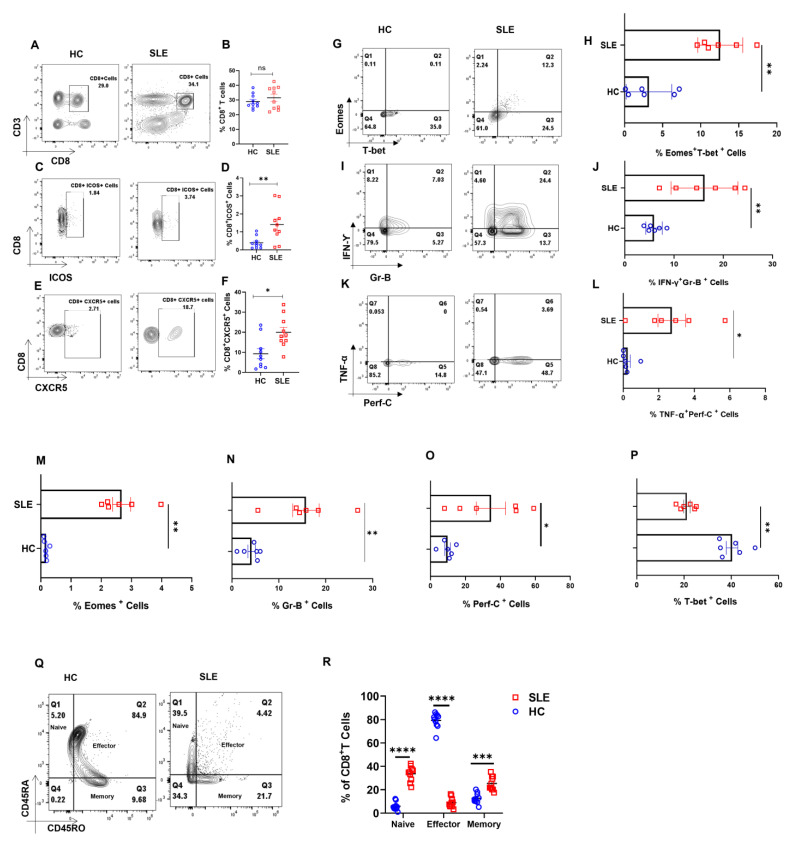
Dysregulated cellular differentiation and expression of activation marker in CD8^+^ T cells of SLE patients. SLE patients display higher activation marker including ICOS and CXCR5 in whole blood. Representative flow cytometry plots show % CD8^+^ T cells in whole blood of healthy controls (*n* = 10) and SLE patients (*n* = 10) (**A**) and cumulative graphical representation (**B**). Representative plots showing % ICOS expression in CD8^+^ T cells from whole blood of HC (*n* = 10) and SLE patients (*n* = 10) (**C**) and cumulative graphical representation (**D**). Representative plots showing expression of % CXCR5 expression on CD8^+^ T cells from whole blood of HC (*n* = 10) and SLE patients (*n* = 10) (**E**) and cumulative graphical representation (**F**). SLE patients (*n* = 6) display higher expression of Eomes and T−bet dual-positive CD8^+^ T cells than healthy controls (*n* = 6) while a reversed trend is observed for T−bet, representative flow cytometry plots (**G**) and cumulative graphical representation (**H**). Representative flow cytometry plots showing increase in inflammatory cytokines IFN−γ, TNF−α and cytotoxic molecules Granzyme−B and Perforin−C in SLE patients (*n* = 6) as compared to healthy control (*n* = 6) (**I**,**K**) and cumulative graphical representation (**J**,**L**). Single positive Eomes (**M**), Gr−B (**N**), Perf−C (**O**) and T−bet (**P**) are also shown from SLE CD8^+^ T cell. Representative plot showing naive, effector, memory compartments between healthy controls (*n* = 10) and SLE patients (*n* = 10) (**Q**) and cumulative graphical representation (**R**). Error bar indicates SEM. Mann–Whitney U test has been used to compare between the groups, and 2-way ANOVA was used to compare two variables among more than two groups. *p* < 0.05 was considered statistically significant (*); *p* < 0.01 was considered to be very significant (**); *p* < 0.001 was considered to be highly significant (***); *p* < 0.0001 was considered extremely significant (****), and ns means non-significant.

**Figure 2 antioxidants-12-00181-f002:**
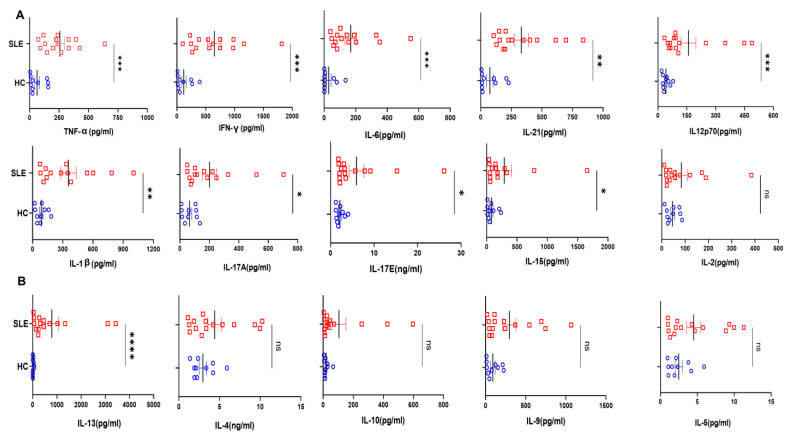
Cytokines analyses in SLE patients and healthy controls. Representative figures from 20 T cell cytokines analyzed in SLE patients’ plasma (*n* = 15) and healthy controls (*n* = 10) are represented as plunge plots. Inflammatory cytokines including IFN−γ, IL−12p40, TNF−α, IL−6, IL−1β, IL−17A, IL−17E, IL−21, IL−15 and IL−21 were significantly elevated in SLE patients as compared to HC (**A**). Among anti-inflammatory cytokines, only IL−13 was elevated in SLE patients, while all others, including IL−10, IL−4, IL−5 and IL−9, did not show any difference between the two groups (**B**). Error bar indicates SEM. Mann–Whitney U Test was performed to compare between the two groups; *p* < 0.05 was considered statistically significant (*); *p* < 0.01 was considered to be very significant (**); *p* < 0.001 was considered to be highly significant (***); *p* < 0.0001 was considered extremely significant (****), and ns means non-significant.

**Figure 3 antioxidants-12-00181-f003:**
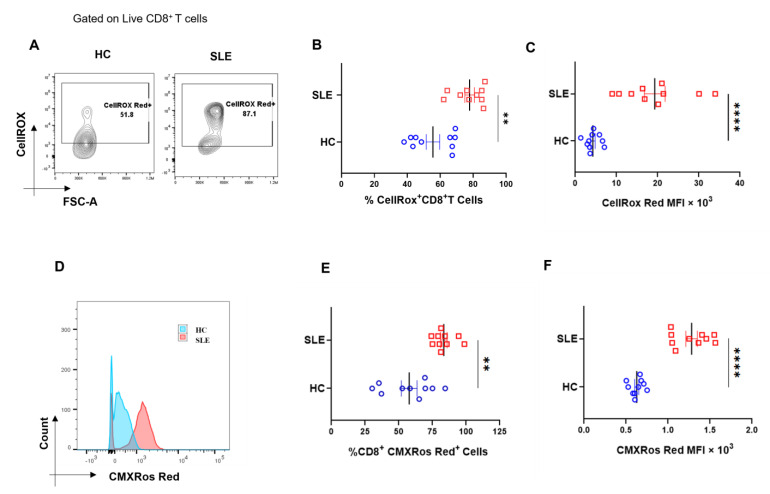
Cellular ROS (cROS) and mitochondrial hyperpolarization in SLE CD8^+^ T cells. CD8^+^ T cells from SLE displayed higher cellular ROS than healthy controls. Representative flow cytometry plots showing CD8^+^ T cells from HC (*n* = 10) and SLE (*n* = 10) (**A**) stained with CellROX Red to determine cellular ROS in CD8^+^ T cells and cumulative graphical representation of the % cells expressing CellROX (**B**). Graphical representation showing difference in the median fluorescence intensity (MFI) of CellROX between HC and SLE (**C**). Representative flow cytometry plots showing CD8^+^ T cells from HC (*n* = 10) and SLE (*n* = 10) stained with CMXROS Red to determine mitochondrial hyperpolarization (**D**). Cumulative graphical representation of the % cells expressing CMXROS Red (**E**) and graphical representation showing difference in the median fluorescence intensity (MFI) of CMXROS Red between HC and SLE (**F**). Error bar indicates SEM. Mann–Whitney U test has been used to compare the groups. *p* < 0.01 was considered to be very significant (**); *p* < 0.0001 was considered extremely significant (****).

**Figure 4 antioxidants-12-00181-f004:**
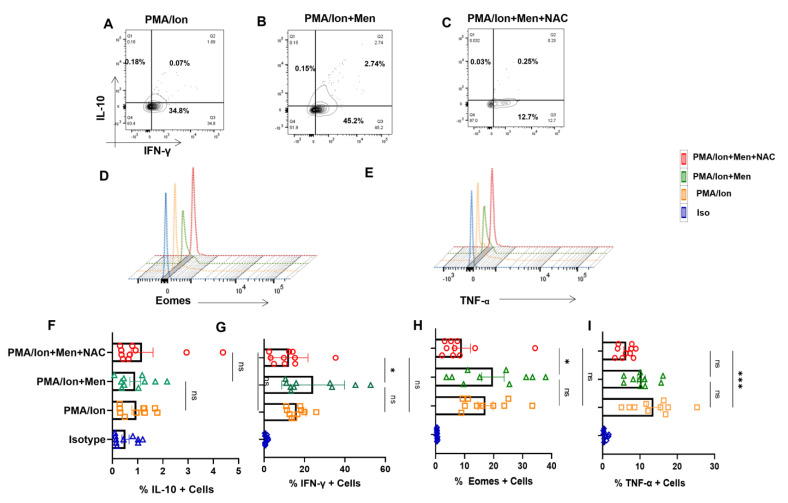
Redox modulation of inflammatory cytokines and transcription factor from SLE CD8^+^ T cells. CD8^+^ T cells were isolated from the PBMCs of SLE patients and treated with 2.5 μM menadione with or without NAC for 4 hrs and processed according to the protocol mentioned in the method section. Representative figures showing reduction in IFN−γ, TNF−α and Eomes following NAC (5 mM) treatment compared to their activation by PMA/Ion or treatment with menadione (2.5 μM) (**A**–**E**). Cumulative graphical representation showing changes in IL−10 and IFN−γ after menadione and NAC treatment (**F**,**G**). Cumulative graphical representation showing changes in Eomes and TNF−α after menadione and NAC treatment (**H**,**I**). Error bar indicates SEM. The experiments were done from 20 SLE patients (*n* = 10) for cytokine assay and (*n* = 10) for transcription factor assay, and all samples were tested for significance using one-way ANOVA, followed by Tukey’s multiple comparison. *p* < 0.05 was considered statistically significant (*); *p* < 0.001 was considered to highly significant (***); and ns means non-significant.

**Figure 5 antioxidants-12-00181-f005:**
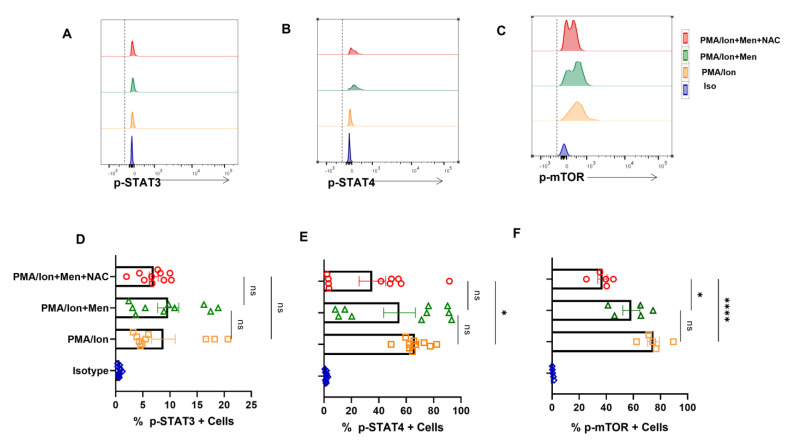
Antioxidant alters expression of p−STAT4 and p−mTOR from SLE CD8^+^ T cells. CD8^+^ T cells were isolated from the PBMCs of SLE patients and treated with 2.5 μM menadione with or without NAC for 4 h and processed according to the protocol mentioned in the method section. Representative figure showing no modulation in p−STAT3 expression when treated with PMA/Ion, menadione (2.5 μM) or NAC (5 Mm) (**A**). Representative figures showing a reduction in p−STAT4 and p−mTOR following NAC (5 mM) treatment compared to their activation by PMA/Ion or treatment with menadione (2.5 μM) (**B**,**C**). Cumulative graphical representation showing changes in p−STAT3, p−STAT4 and p−mTOR after menadione and NAC treatment (**D**–**F**). Error bar indicates SEM. The experiments were done from 15 SLE patients (*n* = 10) for p-STAT assay and (*n* = 5) for p−mTOR assay, and all samples were tested for significance by using one-way ANOVA, followed by Tukey’s multiple comparison. *p* < 0.05 was considered statistically significant (*); *p* < 0.0001 was considered extremely significant (****), and ns means non-significant.

**Figure 6 antioxidants-12-00181-f006:**
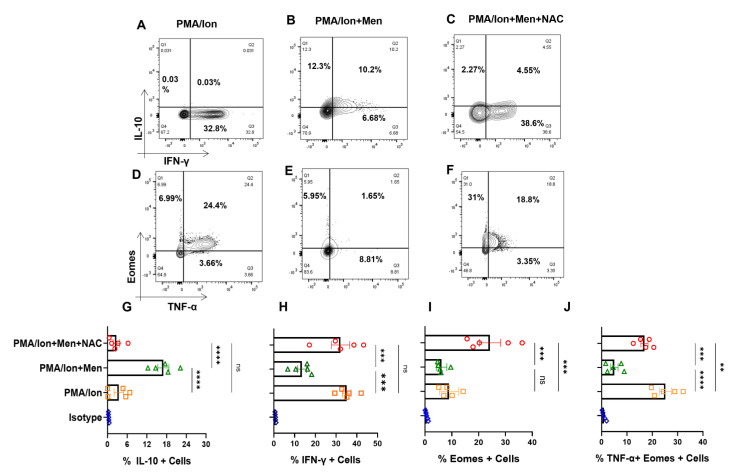
Tc1 cell cytokines are redox-sensitive. CD8^+^ T cells were isolated from PBMCs and differentiated for 10 days following which they are reactivated with PMA/Ion for 4 h and stained with desired antibodies as described in the materials and methods section. Representative figures showing reduction in IFN−γ and increase in IL−10 production following menadione (2.5 μM) treatment compared to their activation by PMA/Ion, which was again restored following treatment with 5 mM NAC (**A**–**C**). Representative figures showing reduction in TNF-α and Eomes expression following menadione (2.5 μM) treatment as compared to their activation by PMA/Ion, which is again restored following treatment with 5 mM NAC (**D**–**F**). Cumulative graphical representation showing significant increase in IL−10 following ROS induction (**G**). Cumulative graphical representation showing significant decrease in IFN−γ following ROS induction (**H**). Cumulative graphical representation showing decrease in the expression of Eomes with menadione treatment, which was restored by subsequent NAC treatment (**I**). Cumulative graphical representation showing reduction in Eomes, TNF−α double-positive Tc1 cells with menadione treatment, which was restored by subsequent NAC treatment (**J**). Error bar indicates SEM. The experiments were performed with five healthy donors, and all of the samples were tested for significance by using one−way ANOVA followed by Tukey’s multiple comparison. *p* < 0.01 was considered to be very significant (**); *p* < 0.001 was considered to highly significant (***); *p* < 0.0001 was considered extremely significant (****), and ns means non-significant.

**Figure 7 antioxidants-12-00181-f007:**
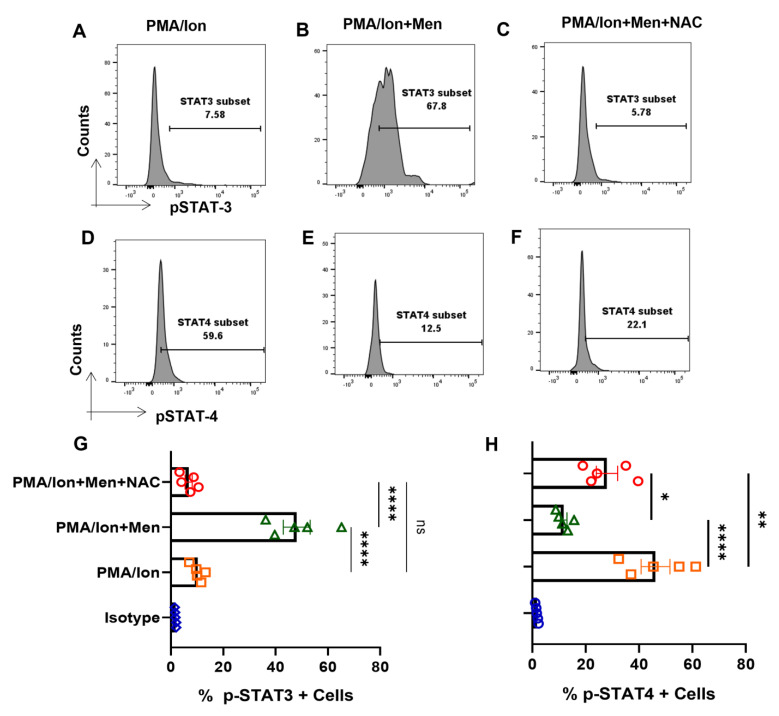
ROS modulates p−STAT’s from terminally differentiated Tc1 cells. Tc1 cells were stimulated with PMA/Ion and treated with 2.5 μM menadione with or without NAC for 3 h and processed according to the protocol mentioned in the method section. Representative figures show increase in p−STAT3 and decrease in p−STAT4, following induction of ROS through menadione compared to their activation by PMA/Ion. Furthermore, following treatment with ROS scavenger NAC, the result is reversed (**A**–**F**). Cumulative graphical data showing significant increase in p−TAT3 expression following ROS induction and their restoration through NAC (**G**). Cumulative graphical data showing significant increase in p−STAT4 expression following ROS induction and their restoration through NAC (**H**). Error bar indicates SEM. The experiments were done from five healthy donors, and all samples were tested for significance by using one-way ANOVA, followed by Tukey’s multiple comparison. *p* < 0.05 was considered statistically significant (*); *p* < 0.01 was considered to be significant; *p* < 0.01 was considered to be very significant (**); *p* < 0.0001 was considered extremely significant (****), and ns means non-significant.

**Figure 8 antioxidants-12-00181-f008:**
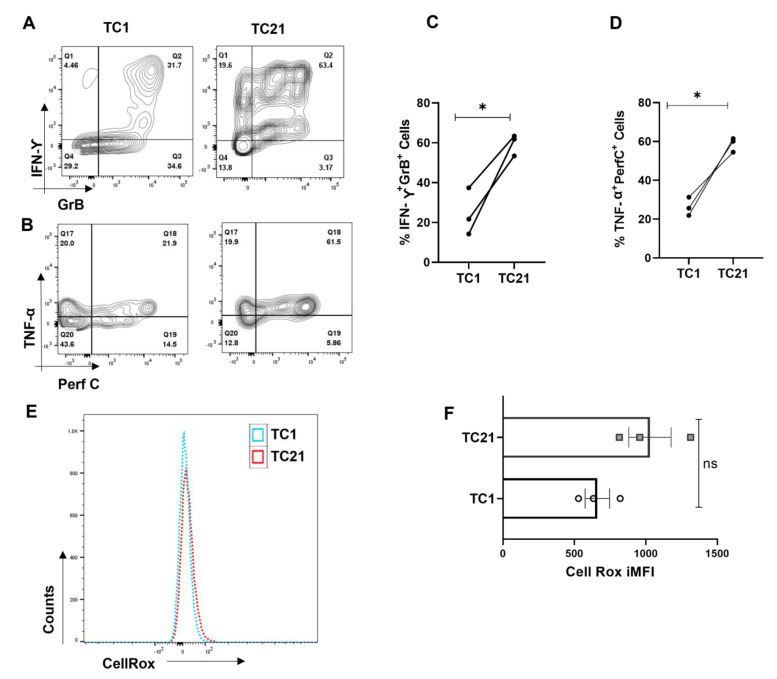
IL-21 drives hyper-cytotoxicity in Tc1 cells. CD8^+^ T cells were isolated from healthy controls (*n* = 3) and cultured for 10 days with and without IL-21 and assessed for different cytokines. Representative flow cytometry figures showing increase in dual positive IFN−γ and Granzyme−B and dual positive TNF−α and Perf−C when IL−21 is added to the Tc1 culture (**A**,**B**) and cumulative graphical data showing the same (**C**,**D**). Representative figure showing cellular ROS in Tc1 and Tc21 (**E**) and cumulative graphical data showing the same (**F**). Error bar indicates SEM. All samples were tested for significance by paired t test (two-tailed) for Tc1 and Tc21cells. Mann–Whitney U test has been used to compare the groups. *p* < 0.05 was considered statistically significant (*); ns means non-significant.

**Table 1 antioxidants-12-00181-t001:** Clinical and demographic characteristics of SLE patients and healthy controls.

	HC (*n* = 26)	SLE (*n* = 41)
Age, years, median (IQR)	28 (25–34)	25 (23–32)
Sex, female/male, *n* (%)	21 (80.76%)/5 (19.23%)	40 (97.5%)/1 (2.4%)
SLEDAI, median (IQR)	N.A	5 (3–6)
Anti-dsDNA, ELISA, U/mL, median (IQR)	N.D	35.28 (10.89–74.72)
C3, g/L, median (IQR)	N.D	0.96 (0.64–1.25)
C4, g/L, median (IQR)	N.D	0.2 (0.19–0.25)
CRP, mg/L, median (IQR)	N.D	7.57 (5.78–16.24)
History of nephritis	N.A	5
Current drug use		
Prednisolone, *n*/*n*, %	N.A	41/41, 100%
Hydroxychloroquine, *n*/*n*, %	N.A	41/41, 100%
Azathioprine, *n*/*n*, %	N.A	14/41, 34.16%
Mycophenolate Mofetil, *n*/*n*, %	N.A	24/41, 58.53%
Methotrexate, *n*/*n*, %	N.A	12/41, 29.26%

SLEDAI-SLE disease activity index, CRP-C-reactive protein, C3-complement component 3, C4-complement component 4, N.D-not determined-not applicable, HC-healthy control, SLE-systemic lupus erythematosus.

## Data Availability

All of the data is contained within the article and the Appendix A.

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
