# Peer review of "IL-21, Inflammatory Cytokines and Hyperpolarized CD8^+^ T Cells Are Central Players in Lupus Immune Pathology"

_antioxidants, 2023, doi:10.3390/antiox12010181_

Round 1
Reviewer 1 Report
I considered the manuscript entitled “IL-21, inflammatory cytokines and hyper polarized CD8+ T cells are central players in Lupus immune pathology” by Soumya Sengupta, that is intended to be published in Antioxidants journal.
I enjoyed the study which appears very well performed, with concise results. I have two concerns. By one hand I claim for a graphical abstract which illustrates the cellular, cytokines and intracellular mechanisms.
On the other hand, I wonder which type of patients the authors had studied. The low SLEDAI, the normal anti DNA titers, CRP and complement suggest patients with very low LES activity. The low number of nephritis patients also may indicate low risk patients. And finally, all of them are under immunosuppressive treatment.
Assuming all these items, then the effects and results of CD8, what means? There is a residual immunological LES activity? It appears that when comparing HC with LES patients some differences may appear. But what would be the situation in a patent with LES outbreak and naïve for any treatment? Please discuss.
Reviewer 2 Report
The manuscript by Sengupta et al. examines CD8 T cell subset in patients with systemic lupus erythematosus (SLE) focusing on the inflammatory phenotype, the oxidant status and the differentiation in cytotoxic T cells. Moreover, the authors investigate the impact of cytokine milieu on CD8 T cells, with special attention for IL-21.
The topic of the article is interesting given that the role of CD8 T cells in SLE is still unclear.
The following points need to be addressed:
- Given that IL-21 is a crucial cytokine in SLE, I suggest to better introduce and describe this cytokine in the Introduction section. The following papers could be useful: PMID:29161463; PMID:17581589; PMID:30180771.
- The discussion section is too long and resembles the Results section. Authors should comment the results by reference to similar previous papers or to potential therapeutic applications.
- At page 7, lines 245-247, the cytokine IL-21 should be included given that it is important for the study and its levels are shown in the figure 2A.
- In the figure 4 (D, E) the significant decrease in the expression of Eomes and TNF-α in NAC treated SLE CD8+ T cells compared with Menadione treated cells and with the PMA/Ion treated population is not visible. Authors could try to display histogram overlay dot plots with the stagger offset to create a psuedo 3-D look.
- Careful English revision is required.
Reviewer 3 Report
The authors performed a series of experiments in order to define some of the aberrant inflammatory and immune responses in SLE. The experiments and the interpretation of the results look sound - however, as I am a clinician, I recommend the review be done by someone with expertise in cellular and molecular immunology / laboratory.
Methods, Statistics: SEM means standard error of the mean, and not standard deviation.
Design: comparison with healthy controls can exaggerate the differences. Future experiments could compare patients with different levels of disease activity/ severity, and maybe with controls with other autoimmune/inflammatory diseases.
Author Response
Please seethe attachment

Round 2
Reviewer 1 Report
I do not understand the ethical concerns with acute patients, but after all the study is ok